# Crop growth dynamics: Fast automatic analysis of LiDAR images in field-plot experiments by specialized software ALFA

**Tadeáš Fryčák**[1], **Tomáš Fürst**[1], **Radoslav Koprna**[2], **Zdeněk Špíšek**[2], **Jakub Miřijovský**[3], **Jan F. Humplík**[2]\*

**1** Department of Mathematical Analysis and Applications of Mathematics, Faculty of Science, Palacký University, Olomouc, Czech Republic, **2** Department of Chemical Biology, Faculty of Science, Palacký University, Olomouc, Czech Republic, **3** Department of Geoinformatics, Faculty of Science, Palacký University, Olomouc, Czech Republic

\* jan.humplik@upol.cz

**Data Availability Statement:** Software tool is freely accessible here: https://github.com/PalackyUniversity/alfa The datasets analysed in the current study are available in public stable data

## Abstract

Repeated measurements of crop height to observe plant growth dynamics in real field conditions represent a challenging task. Although there are ways to collect data using sensors on UAV systems, proper data processing and analysis are the key to reliable results. As there is need for specialized software solutions for agricultural research and breeding purposes, we present here a fast algorithm ALFA for the processing of UAV LiDAR derived point-clouds to extract the information on crop height at many individual cereal field-plots at multiple time points. Seven scanning flights were performed over 3 blocks of experimental barley field plots between April and June 2021. Resulting point-clouds were processed by the new algorithm ALFA. The software converts point-cloud data into a digital image and extracts the traits of interest–the median crop height at individual field plots. The entire analysis of 144 field plots of dimension 80 x 33 meters measured at 7 time points (approx. 100 million LiDAR points) takes about 3 minutes at a standard PC. The Root Mean Square Deviation of the software-computed crop height from the manual measurement is 5.7 cm. Logistic growth model is fitted to the measured data by means of nonlinear regression. Three different ways of crop-height data visualization are provided by the software to enable further analysis of the variability in growth parameters. We show that the presented software solution is a fast and reliable tool for automatic extraction of plant height from LiDAR images of individual field-plots. We offer this tool freely to the scientific community for non-commercial use.

## Introduction

Intensity of plant growth provides key information about the plant's ability to withstand all possible life situations in various environments. Analysis of plant growth is especially useful when performing agricultural experiments and plant breeding programs [1]. However, manual measurements are very laborious, and widely affected by inter-rater variability [2]. Small field-

repository Zenodo under DOI 10.5281/zenodo.10161951 (https://zenodo.org/doi/10.5281/zenodo.10161951).

**Funding:** The work was supported by the European Regional Development Fund project "Plants as a tool for sustainable global development" (No. CZ.02.1.01/0.0/0.0/16_019/0000827). The funder had no role in study design, data collection and analysis, decision to publish, or preparation of the manuscript.

**Competing interests:** The authors have declared that no competing interests exist.

plots that consist of a single treatment (genotype/variety) are often used as the basic unit in plant and agricultural research. To analyze growth (expressed as the change in the crop height in time), the crop height in individual field-plots is manually assessed by some "ruler" resulting in each unit (field-plot) being represented by a single number (or a few numbers). These numbers are taken to represent the crop height over the entire field-plot, regardless of possible spatial heterogeneity which is often observed in agricultural experiments [3]. To obtain more reliable growth data, analysis of crop height in the entire plot area can be performed [4]. This assures that variability in crop height formed by thousands of individual plants will be properly reflected in the results. Moreover, when plant growth is measured, several measurements of crop height are needed to cover most of the vegetative period. This advocates for automatic (e.g. UAV-aided) rather than manual approaches.

For these reasons, several remote-sensing approaches have been developed, such as structure-from-motion (SFM) or LiDAR-based solution. SFM techniques are cheaper but highly demanding from the data processing point of view [5]. On the contrary, LiDAR systems are precise and can be analyzed relatively fast however [6]. There are several limitations that may affect implementation of UAV-LiDAR system to the research workflow. First is the cost of the equipment, then the need to learn how to operate the UAV equipment and to comply with the relevant legislative provisions for its operation. Although the cost of UAV LiDAR systems was very high until recently, there are now systems on the market that rival more advanced RGB orthophotography systems in price. As LiDAR systems provide more accurate data and their processing has significantly lower computational requirements [7], their importance in agricultural research and plant breeding will increase. LiDAR solutions are not limited to airborne systems only, they also include terrestrial LiDARs [8], LiDARs on various moveable platforms such as tractors or sprayers [9]. Both SFM and LiDAR solutions are suitable for generation of crop height models (CHM) derived from point-cloud files. The term CHM was invented for digital surface models (DSM) specifically applied to crops. CHM is defined as a normalized DSM (nDSM) which results from the subtraction of a digital elevation model (DEM = "ground" points only) from DSM containing other features, in our case plant canopy [4, 10].

It has been reported that crop-specific CHM perform better than generalized crop models [11]. Studies were focused on segmentation of individual plants as is typical for maize [12, 13] or on segmentation of entire field-trial plots in case of other cereals, sorghum or cotton [14–17]. The conversion of point-clouds to CHM can be done manually using software for digital terrain model processing [16], but specialized algorithms have also been reported [18]. Although the CHM can be efficiently utilized for segmentation of trial field-plots, they have been found insufficient for individual plant segmentation [19]. For these specific purposes other methods based on direct-point segmentation were developed [20]. Other approaches, such as regional growth [21] or voxel-space projection [22], utilize features of individual points from the point-cloud as well the spatial relations between the points. Most recent progress in LiDAR data application and analysis was comprehensively described by Rivera et al. [23]. Gao et al. [12] recently proposed a method combining RGB orthophoto to identify seed points in young seedlings of maize for their later segmentation from LiDAR point-cloud using fuzzy based C-means clustering analysis. To overcome low accuracy of analysis in cotton-canopy height, Xu et al. [24] developed method implementing canopy laser interception compensation mechanism in their model. Our previous algorithm [25] was based on direct-point analysis strategy to segment trial field-plots of winter wheat. However, direct-point analysis is highly demanding on computing time. This reduced the applicability of our previous algorithm in datasets containing multiple point-clouds. Thus, we propose a new software based on the CHM strategy. The proposed solution reduces the computation time for a 1GB sized point-cloud file from several hours to several minutes without compromising precision and

accuracy. In contrast to the previous algorithm, it can be run efficiently on a standard PC or laptop. The aim of this contribution is to describe the algorithm and show its utility in real conditions.

## Material and methods

Field testing was performed at a location at Palacky University experimental field area in Olomouc (49.5750947 N, 17.2843269 E). Field plot experiments were performed on the spring barley variety Francin (Selgen, CZ) during the year 2021. Field-plots were scanned using UAV system Ricopter VUX-1UAV, Riegl GmBh, Austria) that was operated automatically using the UGCS software (SPH Engineering, Latvia) as described previously [25]. The flight altitude was set to 20 m AGL, horizontal speed of flight 4 m/s, scanning line distance 9 m, calculated side overlap (77%, 31 m). The LiDAR sensor was running at the maximum laser pulse rate (550 kHz). The drone data is presented as a 3D point-cloud format, which consists of a list of [x, y, z] coordinates representing the positions of the recorded points First, we convert this format into a 2.5D digital image format, allowing for the utilization of fast digital image processing methods [26]. However, this conversion results in a loss of information because in a digital image, a single z-value is associated with each [x, y] coordinate [27]. Nevertheless, for crop height measurement purposes, this transformation is suitable. Additionally, no measured z-values are deleted; instead, all z-values corresponding to the same [x, y] coordinate are averaged as described below.

### Data filtering and rasterization

The first step involves cleaning the point-cloud data by eliminating any outlier points that fall outside the 99.9% quantile in the x, y, or z coordinates. Subsequently, a uniform rectangular grid is created in the x-y plane, and each 3D point in the point cloud is assigned to the nearest grid pixel. The elevation of each grid pixel is then calculated as the average of all z-coordinates of cloud points assigned to that specific grid point. During this process, the elevation of certain grid points may remain undefined if no cloud points were assigned to them. To address this, a median filter is applied to impute the missing values [28]. Furthermore, the z-coordinates are quantized to optimize the resulting image for a 16-bit depth. After completing these steps, the original 3D point-cloud files are transformed and saved as significantly smaller 16-bit PNG images, while preserving the crop height information. Each PNG file corresponds to a specific drone measurement time. These PNG images can be conveniently processed using standard image processing tools.

### Selection of Region of Interest (ROI)

The initial image from the repeated measurements must be rotated so that the user can define the ROI. A simple Graphical User Interface (GUI) is provided to the user to enable the rotation and ROI selection in the first image (i.e. the image from the first measurement time). Next, all subsequent images (i.e. images from later measurement times) are automatically cropped and rotated to match the selected ROI.

### Instance segmentation algorithm

Subsequently, the ROI undergoes segmentation to identify the individual field plots within the image for later CHM generation. Vertical and horizontal edges separating the field plots are automatically detected by summing the image values vertically and horizontally (Fig 1A).

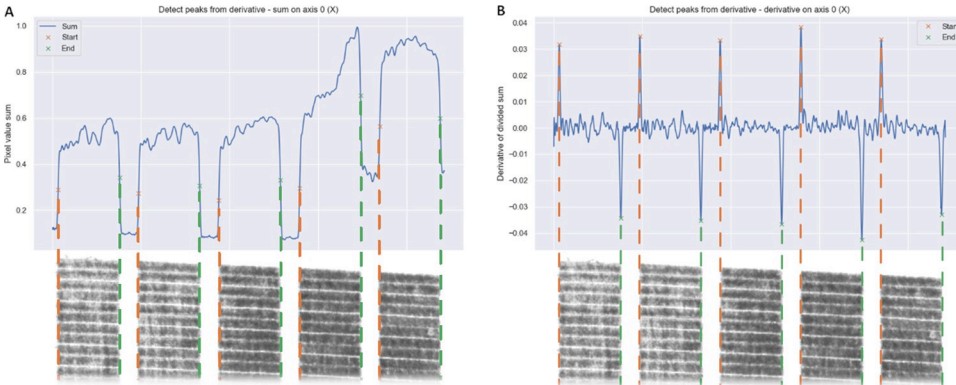

**Fig 1. Instance segmentation of individual fields.** The digital image of a field block highlighted in yellow in Fig 2B is summed along the shorter edge. A) Vertical normalized sums of the processed image (blue) with automatically identified left (green) and right (orange) edges of the individual field plots. B) First derivative of the sum in panel A, enabling the automatic detection of the plot edges.

Peaks are automatically detected in the summed images by means of numerical derivatives (Fig 1B).

## Distortion correction

The individual field plots are often thin and long and their images may be skewed. Thus, we apply a correction by fitting a parallelogram to the boundary of the field plots (Fig 3). The deformation is then automatically rectified through an affine transform algorithm, which is accessible within the OpenCV toolbox [29]. The original image (with distorted plots) is shown in Fig 3A, the result of the affine correction is shown in Fig 3B. Notice that in Fig 3B, the boundaries of the plots are formed by rectangles.

## Crop height computation

The digital image is very "noisy" in the sense that z-values corresponding to neighbouring grid points may be very different. To create a CHM, a maximum filter is used. Each pixel value in the image is substituted by the maximum of its neighbours. Intuitively, this operation corresponds to covering the original image by a deformable blanket. Next, the ground-surface (DEM) at time zero is subtracted from each image to account for any variability in the terrain. For each field-plot and each time of measurement, the following characteristics of the crop height are computed: mean, standard deviation, median, minimum, maximum, and several quantiles. The relative growth rate between time $t_1$ and $t_2$ is computed using the relation [30]

$$RGR(t_1, t_2) = \frac{\ln S(t_2) - \ln S(t_1)}{t_2 - t_1} \qquad (1)$$

Where S(t) is the median crop height at time t.

Each field plot is indexed by its *variant* number $i$ and *block* number $j$ (see Fig 2). The growth in each field-segment is analysed as follows. Let $S_{ij}(t)$ be the median crop height of variant $i$ in block $j$ at time $t$. A logistic growth curve is fitted to the data (separately in each field plot). The logistic curve is defined by:

$$S(t) = \frac{A}{1 + exp(-Bt - C)} \qquad (2)$$

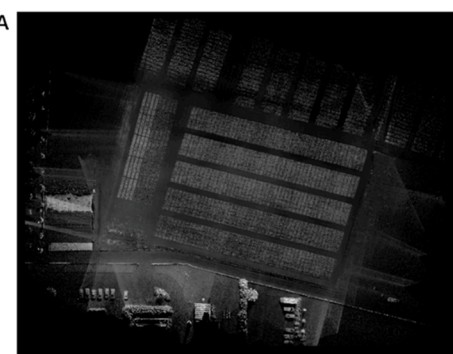
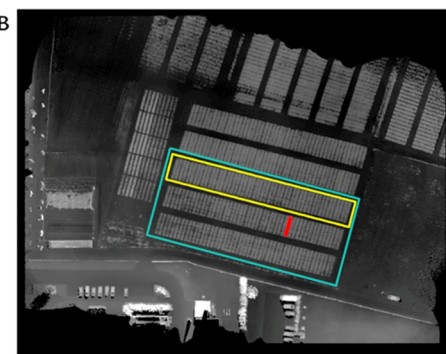

**Fig 2. Initial processing of the point-cloud data into the crop height model.** A) The raw digital image obtained from the point-cloud data by averaging the z-values corresponding to the same [x, y] coordinate. B) The digital image after the application of the median filter to impute the missing values. A single block highlighted in yellow, a single field-plot within the block highlighted in red. The experimental area analysed in the rest of the paper is marked in cyan.

Where $A$ characterizes the maximum height that the crop would reach if observed long enough, $B$ captures the growth velocity, and $C$ captures growth onset (for review see [31]). The parameters $A$, $B$, $C$ are found using a standard non-linear least squares regression (implemented in MATLAB). The variability in the parameters among the field-plots can be visualized by means of a heat map (see Fig 4). The effect of various treatments of the crop can be evaluated by analysing the variability among the A, B and C parameters inferred from the crop heights.

## Results

Data for software testing and optimization were acquired using UAV LiDAR system (Ricopter VUX-SYS, Riegl GmBh, Austria) using same parameters as described previously [25]. Eight flights from March to June were performed to scan an array of barley field-plots organized into three experimental blocks (48 variants per each block, see the cyan experimental area in Fig 2B). LiDAR scans were processed as described previously to point-cloud files in the "las" format [32]. These files were processed by our software to segment point-cloud derived crop height model (CHM) and analyse the crop height at individual field-plots.

To validate the software and assess the UAV LiDAR scanning accuracy, we performed manual measurement using geodetic GPS and a ruler to measure the crop height. These measurements were performed the same day as the UAV LiDAR scanning and covered 6 and 7 field-plots homogeneously distributed across blocks 1 and 2. We performed validation of the crop height measurement in two blocks 1 and 2 by manually measuring the crop height at 5 points

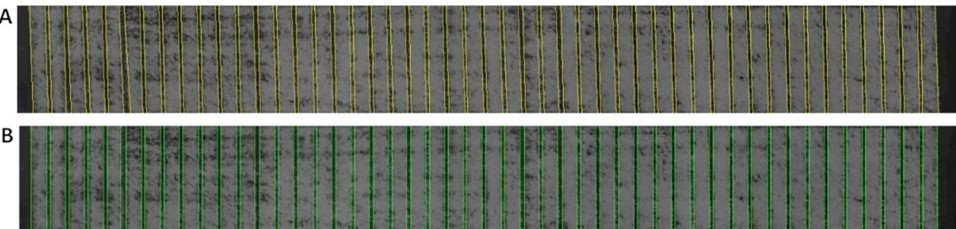

**Fig 3.** Distortion correction A) A detail of a single *block* before the affine correction. Notice the skewed vertical boundaries of the plots. B) The same *block* after the affine correction. The individual plots have now rectangular boundaries.

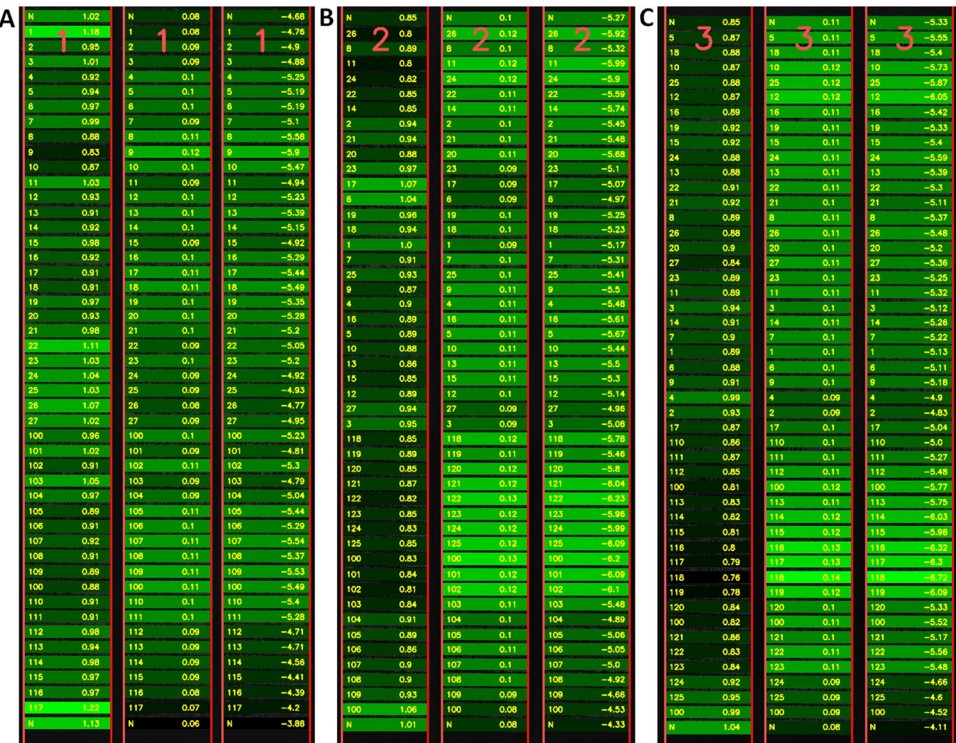

**Fig 4. Visualization of the parameters of the sigmoid growth model for individual field-plots.** Parameter A (left panel) corresponds to the maximum possible height of the crop. Parameter B (middle panel) captures the "growth velocity" because it corresponds to the slope of the sigmoid curve. Parameter C (right panel) is the "offset" which captures the left-right shift of the sigmoid. Parameters are shown for blocks of field-plots 1, 2 and 3 as indicated by the red numbers in the top row. The left column of numbers stands for the ID of the variant, the right column shows the numerical value of the respective parameter of the sigmoid model. The parameter values are color coded. Observe the spatially correlated variability in the parameters.

within each field-plot (see Fig 5). Our validation showed that the Root Mean Square Deviation (RMSD) of the software-computed crop height from the manual measurement was 5.7 cm (0.057 m) for both blocks together. The error distribution for each individual block is depicted in Fig 5.

To assess crop growth dynamics, we measure the crop height at multiple time points during the vegetative developmental phase. In our case, we performed 7 UAV scanning flights over 3 experimental blocks with a total of 144 barley field-plots (see the experimental area marked in cyan in Fig 2). Seven scanning flights were performed between April and June 2021. Scanning was performed in automatic flight mode at 20 m AGL height. After necessary pre-processing steps, the point-cloud files were processed by our software (see Material and Method). Multiple time-point measurements then allow to visualize growth in various ways: A simple bar graph of crop height at particular time-points, relative growth rates based on crop height data, or parameters of the sigmoid model fitted to the growth curve. Plotting crop height (Fig 6A) is the most straight-forward way but it does not allow for easy comparison of growth dynamics in individual field-plots. This is better visualized by a sigmoid fit of the growth curve (Fig 6C). It shows the overall dynamics of vegetative growth for each plot, but information about plant performance from individual time-points is hidden. On the contrary, RGR (Fig 6B) shows the growth rate in particular time-points and helps to understand the overall dynamics of plant growth. The three parameters of the sigmoid (A, B, and C, see section Material and Method above) can also be visualized separately by means of a heat map (see Fig 4).

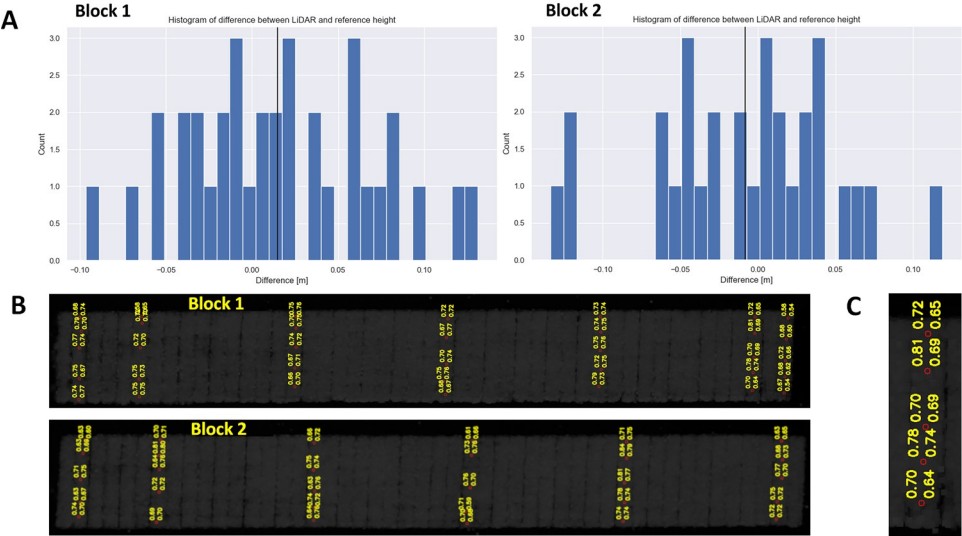

**Fig 5. Manual validation of the ALFA software crop height assessment.** A) Distribution of the difference between the software-computed crop height and the manual measurement. Blocks 1 and 2 are shown separately. B) Visualization of the field-plots that where included in the validation, identification of blocks 1 and 2. C) A detailed view of a selected field-plot showing the points of validation. The values in the right column show the software-computed crop height and the particular point. The values in the left column represent the manual measurement.

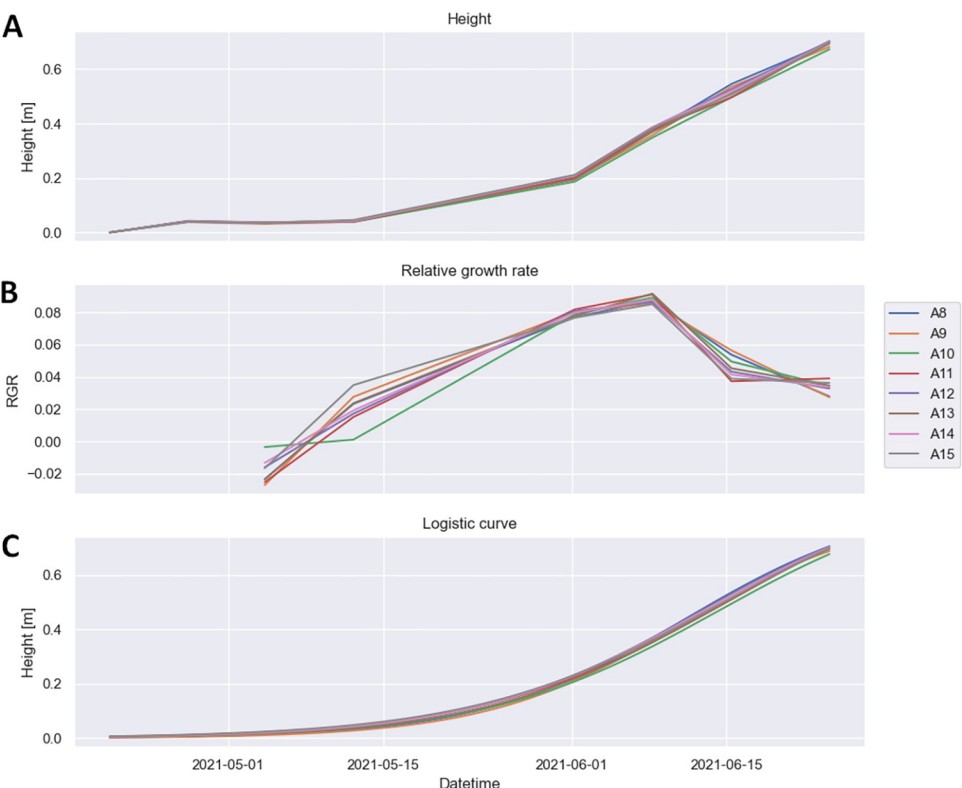

**Fig 6. Barley growth during vegetation season in selected representative field-plots.** For the individual field-plots, the figure shows the evolution of the crop height (A), RGR (B) and logistic (sigmoid) curves (C).

## Discussion

To validate accuracy of LiDAR scanning we validate CHM created by ALFA software by comparison to manual measurements using geodetic GPS with ruler. Root means square error for these measurements was 5.7 cm. This is better than the results reached with the previous software that was validated in wheat field-plots [25]. The manual measurements are based on the median of 5 values, whereas the LiDAR data reflect the height of the entire plot. However, it is hard to say if the validation by few-points based manual measurements relevant to the entire-plot analysis by LiDAR. For these reasons reporting RMSD to manual validation that is common in remote sensing publications is rather indicative parameter than real performance indicator [33].

Plant growth is a dynamic process which is highly non-linear [34]. Classical approach involves studying plant growth at a single, specific point in time. It can provide a snapshot of the plant's state at that moment, including its size, shape, biomass, and other measurable characteristics. This method is relatively simple and requires less time and resources compared to a dynamic analysis. However, it does not capture the changes in plant growth over time and may miss important dynamic processes such are periods of exponential growth or growth stagnation and dormancy (reviewed in [2]).

To test our new software, we evaluated crop growth in 3 blocks, in the total of 144 field-plots. All of the plots contained the same genotype (cv. Francin), sown in a place where we previously observed very heterogeneous growth characteristics in cereals. For this reason, we decided to visualize the growth heterogeneity using a precise measurement of plant growth dynamics. Complexity of plant growth requires multiple-point of views providing better understanding to the researchers. For this reason, numerous mathematical models has been proposed during a time. Beside non-asymptotic linear forms [35, 36] that are not recommended in most cases, various non-linear models were developed. These models better reflect real dynamics of growth affected by various environmental events. Based on plant species their life strategy and growing conditions researchers may choose different non-linear model. Most common is expression of plant growth as three-parameter logistic curve [37–39] that is also case of our software (see Fig 6). Other option is to use Gompertz model that differs from previous one in manifestation of inflection point (for comprehensive review see [40]). However, ALFA software is flexible and growth model and resulting curves can be easily changed in software code. Important differences in growth dynamics are obvious from the RGR and growth curve parameters (Fig 6B and 6C). Although we do not put any biological hypothesis in this methodological study it should be noted that some biological relevant differences manifested in growth parameters. The heterogeneity is particularly visible in the heat map (Fig 4) of the sigmoid growth model parameters. The heat map shows "patterns" of spatially correlated growth differences. It is known that barley is sensitive to different soil compaction that significantly affects root and shoot growth [41]. Since chemical analysis did not reveal any important variability in chemical and nutritional composition of soil, we presume that the physical properties of the soil play the key role.

## Conclusion

The aim of the software development was to design and optimize an algorithm capable of fast extraction of crop height from point-clouds obtained by UAV LiDAR scanning. Recent literature clearly shows that there is lack of specialized software solutions for automatic extraction of plant/plot/canopy features. As showed in systematic review of Rivera et al. [23], most of the published studies applying LiDAR in agriculture reported data processing by some of the generic geoinformatics software with manual point-cloud analysis. In contrast, our software

automatically builds CHM and extract plant height for all experimental field-plots. The ALFA software (available freely for non-commercial use here: https://github.com/PalackyUniversity/alfa) uses point-cloud data, converts them into a digital image, and extracts the trait of interest. The entire analysis of 144 field plots (1.2 x 9 meters each) measured at 7 time points takes about 3 minutes at a standard PC. The RMSD difference between the ALFA software computed crop height (from UAV LIDAR data) and a manual measurement (by a ruler) was 5.7 cm. Three different ways of expressing the plant growth dynamics are introduced to allow for understanding the plant growth dynamics and plant reactions to environmental changes. For the future, we plan to add a statistical toolbox to the ALFA software. The toolbox will assess the plant growth dynamic by means of hierarchical Bayesian model that correctly accounts for the spatially correlation of the growth dynamics. In this way, the true effects of treatment can be extracted from the inherently correlated growth data.

## Author Contributions

**Conceptualization:** Jan F. Humplík.

**Data curation:** Jakub Miřijovský.

**Formal analysis:** Zdeněk Špíšek, Jakub Miřijovský.

**Methodology:** Tomáš Fürst, Radoslav Koprna, Zdeněk Špíšek, Jakub Miřijovský, Jan F. Humplík.

**Software:** Tadeáš Fryčák, Tomáš Fürst, Jakub Miřijovský.

**Supervision:** Radoslav Koprna, Jan F. Humplík.

**Validation:** Zdeněk Špíšek, Jan F. Humplík.

**Writing – original draft:** Tadeáš Fryčák, Tomáš Fürst, Jakub Miřijovský, Jan F. Humplík.

**Writing – review & editing:** Tomáš Fürst, Jan F. Humplík.

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
