## [Decision Letter · Decision Letter 0]

4 Oct 2023

PONE-D-23-28678ALFA - Airborne LiDAR Field-plot Analysis. Fast automatic point-cloud filtering algorithm for analysis of crop growth dynamics in field-plot experiments.PLOS ONE

Dear Dr. Humplik,

Thank you for submitting your manuscript to PLOS ONE. After careful consideration, we feel that it has merit but does not fully meet PLOS ONE’s publication criteria as it currently stands. Therefore, we invite you to submit a revised version of the manuscript that addresses the points raised during the review process.

The manuscript must be corrected in all points indicated by the reviewers, such as:

1. The title should be rewritten.

2. Improve the abstract by adding some your key results.

3. The most recent studies should be included in the literature review section.

4. The description and the presentation of your results should be improved.

5. I can not see the difference before and after the affine correction, clarify it.

6. to provide comprehensive evaluation and recommendations for further study.

7. Include your study's hypothesis.

8. Avoid let your manuscript like a case report.

We look forward to receiving your revised manuscript.

Kind regards,

Claudionor Ribeiro da Silva

Academic Editor

PLOS ONE

Journal Requirements:

 "The work was supported by the ERDF project "Plants as a tool for sustainable global development" (No. CZ.02.1.01/0.0/0.0/16_019/0000827)." 

4. Please expand the acronym “ERDF” (as indicated in your financial disclosure) so that it states the name of your funders in full.

Reviewers' comments:

Reviewer's Responses to Questions

**Comments to the Author**

1. Is the manuscript technically sound, and do the data support the conclusions?

Reviewer #1: Yes

Reviewer #2: Yes

Reviewer #3: Yes

2. Has the statistical analysis been performed appropriately and rigorously? 

Reviewer #1: Yes

Reviewer #2: Yes

Reviewer #3: No

3. Have the authors made all data underlying the findings in their manuscript fully available?

Reviewer #1: Yes

Reviewer #2: Yes

Reviewer #3: Yes

4. Is the manuscript presented in an intelligible fashion and written in standard English?

Reviewer #1: Yes

Reviewer #2: Yes

Reviewer #3: Yes

5. Review Comments to the Author

Reviewer #1: The paper has been meticulously composed, proficiently elucidating the robustness of the employed algorithm, the requisite dataset, and the strategies for effective plant height optimization.

The authors could highlight the limitations and complexities associated with the implementation of the UAV LiDAR system. Additionally, they should outline their intended measures for enhancing user-friendliness, particularly for farmers with fundamental computer skills.

Reviewer #2: 1. The title should be rewritten.

2. Selection of Region of Interest at line 101 should be corrected.

3. Figure 2 is not clear.

4. I can not see the difference before and after the affine correction, clarify it.

5. In line 206,

Heterogeneity is particularly visible in the heat map (Fig 4) of the sigmoid growth model parameters. Discuss and clarify the heat map.

Reviewer #3: The abstract

The authors should focus on improving your abstract by adding some your key results such as

In the line 24

You can add this sentence: The entire analysis of 144 field plots measured at 7 time points takes about 3 minutes at a standard PC. it should be included.

Also, you should add the key finding of validation part.

Please include the research problem in abstract section.

The introduction

I appreciate the introduction. It was well-written. However, in the 3 first sentence needs references.

The introduction is beautifully written. I appreciate it. However, the most recent studies should be included in the literature review section. Please review the most recent publications, such as those published between 2017 and 2022, to identify any research gaps.

2.The materials and methods

Please, enhance the section of materials and methods. I recommended confirm the references of all methods.

Please, you should numbering the equations.

3. Results and Discussion

Please the description of your results and the presentation of your results should be improved.

Expanding your discussion will help your paper flow better. By analyzing your finding, connecting it to recent research, and describing the main results in your discussion, you can further expand your argument

Answering why is this manuscript important?

Greatly improve the mechanistic arguments about dynamical analysis. In general, some mechanistic arguments are not well developed.

4. CONCLUSION

I advise to provide comprehensive evaluation and recommendations for further study, which are required in the paper's conclusion. if the discussion and the findings are kept separate. It will get better.

Rethink to improve the quality of each figure and the design of the table.

Other comments:

Please include your study's hypothesis along with the data statistical analysis.

The statistical analysis can’t satisfy your study because there is no statistical analysis.

The study seems only optimizing an algorithm of fast extraction of crop height. It could hardly attract the international readers interests. I recommend avoiding let your manuscript like a case report.

6. PLOS authors have the option to publish the peer review history of their article (what does this mean?). If published, this will include your full peer review and any attached files.

Reviewer #1: **Yes: **Abdul Aziz Karim

Reviewer #2: **Yes: **Shaimaa Alexeree

Reviewer #3: No

---

## [Author Response · Author response to Decision Letter 0]

20 Nov 2023

Reviewer and editor comments

Editor

1. The title should be rewritten.

Thank you for your valuable comments. The title has been changed according recommendation.

2. Improve the abstract by adding some your key results.

Thank you. We add key results and software features to the abstract.

3. The most recent studies should be included in the literature review section.

The Introduction section has been enriched for most recent references. However, we have tried to keep the Introduction section as straightforward and understandable as possible. Therefore, we refer those interested in a detailed analysis directly to the high-quality comprehensive review by Rivera et al. 2023.

4. The description and the presentation of your results should be improved.

The results and discussion section has been split into self-contained chapters and substantially rewritten. Also, Figures 1 and 2 have been revised as recommended by the reviewers.

5. I can not see the difference before and after the affine correction, clarify it.

The description of Figure 3 has been changed to clearly explain the process of distortion correction. Whereas in raw image the plots do not have a vertical boundaries, affine correction fix this issue to make rectangular shapes.

7. Include your study's hypothesis.

This paper is methodological/software paper, for this reason we do not propose any hypothesis or biological question. On the other hand, publishing this software and make it freely available, may help to test the hypotheses of many researchers worldwide.

8. Avoid let your manuscript like a case report. 

Our study aims mainly to develop and test software for LiDAR data processing that is lacking in biology/agricultural research community as obvious from systematic review of Rivera et al., 2023. For this reason our paper do not describe case study, but design of novel software solution. This may help to the community not only to use the software, but also develop new generation of similar software or adopt it as core of new software solution.

Reviewer #1: 

The paper has been meticulously composed, proficiently elucidating the robustness of the employed algorithm, the requisite dataset, and the strategies for effective plant height optimization. The authors could highlight the limitations and complexities associated with the implementation of the UAV LiDAR system. Additionally, they should outline their intended measures for enhancing user-friendliness, particularly for farmers with fundamental computer skills.

Thank you for your valuable comments. We add commentary to the UAV-LiDAR implementation into Introduction section according your comment. Also we note in the Conclusion section that we plan to develop statistical toolbox that may help to the users analyze their data directly in the software. However, there will be still certain level of computer skill needed to use the software. On the other hand the software is mainly intended for Agricultural Research and Breeding stations, where this level of computer skills may be expected. 

Reviewer #2: 

1. The title should be rewritten.

Thank you for your valuable comments. The title has been changed according recommendation.

2. Selection of Region of Interest at line 101 should be corrected. 

 Thank you. This part has been rewritten according your recommendation. 

3. Figure 2 is not clear.

 Thank you, Figure 2 and its description were revised to provide clear message. Hopefully, it satisfied your comment. 

4. I can not see the difference before and after the affine correction, clarify it.

 The description of Figure 3 has been changed to clearly explain the process of distortion correction. Whereas in raw image the plots do not have a vertical boundaries, affine correction fix this issue to make rectangular shapes.

5. In line 206, heterogeneity is particularly visible in the heat map (Fig 4) of the sigmoid growth model parameters. Discuss and clarify the heat map.

 Thank you. This issue is discussed from the line 279 in Discussion part. We revised this part to clarify the heterogeneity in the field. As stated in Discussion it may be caused by different physical properties in the soil, for which spring barley is very sensitive.

Reviewer #3: 

Manuscript Number: PONE-D-23-28678 

Title: ALFA - Airborne LiDAR Field-plot Analysis. Fast automatic point-cloud filtering algorithm for analysis of 3 crop growth dynamics in field-plot experiments

Dear editor

The manuscript is potentially a publishable study, and is related to the journal. 

But the discussion is not well prepared. The scientific issues of the paper are not focused, mostly data representation, and the discussion part is not well presented. 

Thank you for your valuable comments. The results and discussion section has been split into self-contained chapters and substantially rewritten. Also, Figures 1 and 2 have been revised as recommended by the reviewers. Discussion part was rewritten, new references were added to cover complexity of data analysis mainly from point of view of dynamics of plant growth.

The main objectives of the current study are to extract the information on crop height at many individual cereal field-plots at multiple time points

This subject is crucial for smart agriculture. However, there are some remarks that should be taken into consideration. The discussion must to be supported by earlier researchers and consider any peer review of the findings of previous studies. So, it could hardly attract the international readers interests. 

Thank you. Our software is mainly intended for agricultural research where the field-plot system is common. Specialized software for LiDAR data processing that is lacking in biology/agricultural research community as obvious from systematic review of Rivera et al., 2023. Publishing it in open journal may help to the community not only to use the software, but also develop new generation of similar software or adopt it as core of new software solution for different purposes than described in the paper. For these reasons we believe that it may attract readers interest more easily than some really specialized enclosed study.

The abstract 

The authors should focus on improving your abstract by adding some your key results such as 

 Thank you. We add key results and software features to the abstract.

In the line 24

You can add this sentence: The entire analysis of 144 field plots measured at 7 time points takes about 3 minutes at a standard PC. it should be included. 

 Thank you. It has been included according your recommendation.

 Also, you should add the key finding of validation part.

 Key findings were noted in the Abstract.

Please include the research problem in abstract section.

 Thank you. Main research problem was lacking useful software solution that is noted in the Abstract now.

The introduction

I appreciate the introduction. It was well-written. However, in the 3 first sentence needs references.

Thank you for your comment. We add references according your recommendation.

The introduction is beautifully written. I appreciate it. However, the most recent studies should be included in the literature review section. Please review the most recent publications, such as those published between 2017 and 2022, to identify any research gaps.

The Introduction section has been enriched for most recent references. However, we have tried to keep the Introduction section as straightforward and understandable as possible. Therefore, we refer those interested in a detailed analysis directly to the high-quality comprehensive recent review by Rivera et al. 2023.

2.The materials and methods 

Please, enhance the section of materials and methods. I recommended confirm the references of all methods. 

The original Implementation section has been rearranged, rewritten and renamed Materials and Methods. New literature references have been added.

Please, you should numbering the equations. 

Equations were numbered according your recommendation.

3. Results and Discussion

Please the description of your results and the presentation of your results should be improved.

The results and discussion section has been split into self-contained chapters and substantially rewritten. Also, Figures 1 and 2 have been revised as recommended by the reviewers.

Expanding your discussion will help your paper flow better. By analyzing your finding, connecting it to recent research, and describing the main results in your discussion, you can further expand your argument

Discussion section has been substantially rewritten, new references were added and most important topics are newly discussed. 

Answering why is this manuscript important?

Specialized software for LiDAR data processing is lacking in biology/agricultural research community as obvious from systematic review of Rivera et al., 2023. Publishing it in open journal may help to the community not only to use the software, but also develop new generation of similar software or adopt it as core of new software solution for different purposes than described in the paper. For these reasons we believe that publishing manuscript with our software may importantly affect the research in this field.

Greatly improve the mechanistic arguments about dynamical analysis. In general, some mechanistic arguments are not well developed.

The Discussion part has been importantly rewritten and topic about modeling of plant growth dynamics was added. We believe that this help to the readers to understand better this important problematics.

 4. CONCLUSION 

I advise to provide comprehensive evaluation and recommendations for further study, which are required in the paper's conclusion. if the discussion and the findings are kept separate. It will get better.

Thank you for your comment. As we noted in the conclusion, the most important future activity connected to the ALFA software will be development of statistical toolbox that will help to users to analyze data directly in the software. For this we need to provide software to the wide community and based on their feedback and experiences toolbox will developed and optimized. The results and discussion section has been split into self-contained chapters and substantially rewritten.

Rethink to improve the quality of each figure and the design of the table.

Thank you. The figures 1 and 2 were modified and quality of figures was improved. In this paper we do not present any table. Hopefully it will satisfy your comments.

Other comments: 

Please include your study's hypothesis along with the data statistical analysis. 

Thank you for your comments. This paper is methodological/software paper, for this reason we do not propose any hypothesis or biological question. On the other hand, publishing this software and make it freely available, may help to test the hypotheses of many researchers worldwide.

The statistical analysis can’t satisfy your study because there is no statistical analysis.

Thank you for your comment. We do not provide any statistical analysis, because we do not test any hypothesis in this methodological/software article. However, anybody who need to accurate compare the growth of cereal or similar field-plots will benefit from the manuscript, because she/he can test their hypothesis based on relevant and accurate data from LiDAR scanning. The statistical analysis of logistic curves is really interesting problem that do not posses straightforward solution as described e.g. in Tsoularis and Wallace (2002). As we noted in the conclusion, the most important future activity connected to the ALFA software will be development of statistical toolbox that will help to users to analyze data directly in the software. This toolbox will be based on Bayesian statistics as we did previously for seedling germination (Kaplan-Meier) curves (Humplík et al., 2020; http://www.bayes4plants.com/). 

The study seems only optimizing an algorithm of fast extraction of crop height. It could hardly attract the international readers interests. I recommend avoiding let your manuscript like a case report.

Thank you. Our software is mainly intended for agricultural research where the field-plot system is common. Specialized software for LiDAR data processing that is lacking in biology/agricultural research community as obvious from systematic review of Rivera et al., 2023. Publishing it in open journal may help to the community not only to use the software, but also develop new generation of similar software or adopt it as core of new software solution for different purposes than described in the paper. For these reasons we believe that it may attract readers interest more easily than some really specialized enclosed study.

Tsoularis, A., & Wallace, J. (2002). Analysis of logistic growth models. Mathematical biosciences, 179(1), 21-55.

---

## [Decision Letter · Decision Letter 1]

29 Dec 2023

Crop growth dynamics: Fast automatic analysis of LiDAR images in field-plot experiments by specialized software ALFA.

PONE-D-23-28678R1

Dear Dr. Humplik,

We’re pleased to inform you that your manuscript has been judged scientifically suitable for publication and will be formally accepted for publication once it meets all outstanding technical requirements.

Kind regards,

Claudionor Ribeiro da Silva

Academic Editor

PLOS ONE

Additional Editor Comments (optional):

Reviewers' comments:

Reviewer's Responses to Questions

**Comments to the Author**

1. If the authors have adequately addressed your comments raised in a previous round of review and you feel that this manuscript is now acceptable for publication, you may indicate that here to bypass the “Comments to the Author” section, enter your conflict of interest statement in the “Confidential to Editor” section, and submit your "Accept" recommendation.

Reviewer #1: All comments have been addressed

Reviewer #2: All comments have been addressed

2. Is the manuscript technically sound, and do the data support the conclusions?

Reviewer #1: Yes

Reviewer #2: Yes

3. Has the statistical analysis been performed appropriately and rigorously? 

Reviewer #1: Yes

Reviewer #2: Yes

4. Have the authors made all data underlying the findings in their manuscript fully available?

Reviewer #1: Yes

Reviewer #2: Yes

5. Is the manuscript presented in an intelligible fashion and written in standard English?

Reviewer #1: Yes

Reviewer #2: Yes

6. Review Comments to the Author

Reviewer #1: (No Response)

Reviewer #2: (No Response)

7. PLOS authors have the option to publish the peer review history of their article (what does this mean?). If published, this will include your full peer review and any attached files.

Reviewer #1: No

Reviewer #2: **Yes: **Shaimaa Alexeree

---

## [Editor Report · Acceptance letter]

9 Jan 2024

PONE-D-23-28678R1 

PLOS ONE

Dear Dr. Humplik, 

I'm pleased to inform you that your manuscript has been deemed suitable for publication in PLOS ONE. Congratulations! Your manuscript is now being handed over to our production team.

Kind regards, 

on behalf of

Dr. Claudionor Ribeiro da Silva 

Academic Editor

PLOS ONE